# The Emerging Role of MicroRNAs in Regulating the Drug Response of Cholangiocarcinoma

**DOI:** 10.3390/biom10101396

**Published:** 2020-09-30

**Authors:** Wen-Kuan Huang, Chun-Nan Yeh

**Affiliations:** 1Division of Hematology-Oncology, Department of Internal Medicine, Chang Gung Memorial Hospital at Linkou, Chang Gung University College of Medicine, Taoyuan 333, Taiwan; medfoxtaiwan@gmail.com; 2Department of Surgery and Liver Research Center, Chang Gung Memorial Hospital at Linkou, Chang Gung University College of Medicine, Taoyuan 333, Taiwan

**Keywords:** cholangiocarcinoma, microRNAs, drug resistance, gemcitabine, cisplartin, chemotherapy, targeted therapy

## Abstract

Cholangiocarcinoma (CCA) is the most common biliary malignancy, and has a poor prognosis. The median overall survival with the standard-of-care chemotherapy (Gemcitabine and cisplatin) in patients with advanced-stage CCA is less than one year. The limited efficacy of chemotherapy or targeted therapy remains a major obstacle to improving survival. The mechanisms involved in drug resistance are complex. Research efforts focusing on the distinct molecular mechanisms underlying drug resistance should prompt the development of treatment strategies that overcome chemoresistance or targeted drug resistance. MicroRNAs (miRNAs) are a class of evolutionarily conserved, short noncoding RNAs regulating gene expression at the post-transcriptional level. Dysregulated miRNAs have been shown to participate in almost all CCA hallmarks, including cell proliferation, migration and invasion, apoptosis, and the epithelial-to-mesenchymal transition. Emerging evidence demonstrates that miRNAs play a role in regulating responses to chemotherapy and targeted therapy. Herein, we present an overview of the current knowledge on the miRNA-mediated regulatory mechanisms underlying drug resistance among CCA. We also discuss the application of miRNA-based therapeutics to CCA, providing the basis for innovative treatment approaches.

## 1. Introduction

Cholangiocarcinoma (CCA), arising from the epithelial cells of the intra- (5‒10%) and extra-hepatic bile ducts (90‒95%), is the most common primary biliary malignancy, with increasing global incidence rates [1]. While surgical resection with curative intent is the only mainstay treatment for resectable CCA, more than two-thirds of patients are not eligible for surgical resection due to advanced stage at presentation. Unresectable CCA carries a poor prognosis, with a median survival of 11.7 months [2]. Cisplatin and gemcitabine combination chemotherapy is the current standard first-line treatment for patients with advanced CCA. 5-fluorouracil-based regimens have also been recommended in the National Comprehensive Cancer Network (NCCN) guidelines [3]. Gemcitabine-based regimens may be superior to 5-fluorouracil-based regimens for improving overall survival [4]. However, the antitumor activity of chemotherapy is far from satisfactory, encouraging further clinical trials of novel molecularly-targeted therapies.

Comprehensive whole-exome sequencing and transcriptome analysis has revealed that each CCA subtype has distinct genetic alterations [5], indicating the marked genotypic heterogeneity of CCA. Moreover, molecular profiling suggests that grouping together CCA subtypes in a clinical trial rather than stratifying patients based on molecular pathogenesis may lead to misinterpretation of treatment efficacy. To date, there is no FDA-approved targeted therapy for advanced CCA. Previous clinical trials have shown disappointing results for gemcitabine-based chemotherapy in combination with an epithelial growth factor receptor (EGFR) inhibitor such as cetuximab and erlotinib [6,7]. Some therapeutic agents, targeting fibroblast growth factor receptor (FGFR) 2, isocitrate dehydrogenase (IDH) 1/2, and the sphingosine kinase pathway, have demonstrated encouraging results [8,9,10,11], holding promise for future randomized biomarker-driven studies. Moreover, the potential benefit of immune checkpoint blockade with antiprogrammed death (PD)-1 antibody in advanced CCA is currently being explored. While preliminary studies have not shown promising effects in PD-L1-positive CCA, a subsequent phase II study demonstrated an overall response rate of 40.9%, with an average survival of 24.3 months in previously treated patients with microsatellite instability-high (MSI-H)/mismatch repair-deficient (dMMR) advanced CCA [12]. Tumors with MSI-H/dMMR carry a high mutational burden and the potential for increased neoantigen load, eliciting a response to anti-PD-1 antibody immunotherapy. Although only 5‒10% of CCA has MSI-H/dMMR, these results pave the way for the evolution of the treatment paradigm, shifting from a one-size-fits-all approach to precision or personalized treatment.

Drug resistance is a common phenomenon limiting efficacy in cancers, including CCA. Of note, the prominent features of CCA stroma are desmoplastic and hypovasculuzed, which limit the therapeutic efficacy and promote tumor progression. Gemcitabine-based doublet chemotherapy for CCA has achieved an overall response rate of 20‒30%, and a median progression-free survival (PFS) of 5‒8.5 months [13]. Pemigatinib, a selective inhibitor of FGFR, achieved an objective response of 36% and a median PFS of 6.9 months in previously treated patients with intrahepatic CCA who have FGFR2 fusions or rearrangements [11]. Altogether, only subsets of patients respond, and the duration of tumor response, when present, is not sustained beyond a few months, indicating that the inevitable development of drug resistance remains a major obstacle hindering treatment outcomes.

MicroRNAs (miRNAs), first discovered by Ambros and colleagues in 1993 [14], are a class of endogenous noncoding, single-stranded, small RNA molecules, with a length of 18‒25 nucleotides. MiRNAs regulate gene expression at the post-transcriptional level through binding to the seed sequence in the 3’-untranslated region of target messenger RNA, leading to degradation or translation repression [15]. MiRNAs play a crucial role in regulating fundamental cellular processes, including cell proliferation, differentiation, migration, cell cycle, metabolism, and apoptosis, which are often disturbed in cancer [16]. The first report of miRNA profiling in human intrahepatic CCA revealed two distinct clusters closely associated with the expression levels of carbohydrate antigen 19-9 [17]. Further studies demonstrated the functional roles of dysregulated miRNAs in the development of CCA, suggesting that some miRNAs can act as oncomiRs or tumor suppressor miRNAs, based on their targeted genes [18,19]. Moreover, miRNAs from tumor tissue or blood may serve as diagnostic or prognostic biomarkers for CCA [20]. Increasing evidence suggests that aberrant expression of several miRNAs is involved in modulating the response to chemotherapy or other targeted treatments (Table 1). Moreover, combined miRNA-based therapeutics, to improve the drug resistance, is a promising treatment strategy. Thus, in the following paragraphs, we summarize the literature on miRNAs linked to drug resistance in CCA, and the recent advances in miRNA-based therapeutics in CCA.

## 2. MiRNAs and Resistance to Gemcitabine

Gemcitabine, a nucleoside analogue of cytidine, mediates cytotoxicity by inducing DNA strand breaks and inhibiting DNA repair through abolishing ribonucleotide reductase. Several miRNAs have been reported to be involved in modulating gemcitabine response or resistance in CCA. The mechanisms underlying the role of miRNAs in gemcitabine resistance are summarized in Figure 1.

### 2.1. miR-21

As one of the first miRNAs discovered in the human genome, miR-21 is a major oncomir, with overexpression in several gastrointestinal cancers, including hepatocellular carcinoma [35], cholangiocarcinoma [21], pancreatic [36], and esophageal cancer [37]. Upregulated miR-21 has been shown to promote tumor growth, invasion, and epithelial mesenchymal transition (EMT) in cholangiocarcinoma [21]. For example, three tumor suppressors, including programmed cell death 4 (PDCD4) [38], phosphatase, and tensin homolog (PTEN) [38], and reversion-inducing cysteine-rich protein with Kazal motifs (RECK) [39], have been shown to be regulated by miR-21 in CCA. By modulating miR-21 expression by ectopic overexpression or silencing of miR-21 in Mz-ChA-1 cells, Meng et al. demonstrated that PTEN and phosphoinositide-3-kinase (PI3K)/AKT signaling are regulated by miR-21. MiR-21 overexpression decreased gemcitabine-induced apoptosis. This decreased cytotoxic effect was rescued by adding an AKT inhibitor or mTOR inhibitor. These findings indicate that miR-21 represses gemcitabine cytotoxicity by targeting PTEN, resulting in an attenuation of its inhibition on AKT/mTOR signaling [21].

### 2.2. miR-141

As a member of the miR-200 family, miR-141 has been found to serve as a tumor suppressor in several gastrointestinal cancers [40,41]. However, miR-141 was found to be highly expressed in CCA cell lines, which may be associated with the frequent gain of the chromosomal site of miR-141 (12p13.31) [42]. Upregulated miR-141 enhanced malignant cholangiocyte growth. In contrast, inhibition of miR-141 reduced the growth of CCA cell lines [21]. These findings suggest that miR-141 acts as an oncomiR to promote cholangiocarcinoma proliferation in CCA. 

Ectopic overexpression of miR-141 increased the viability of cholangiocytes in response to gemcitabine. Moreover, the silencing of miR-141 expression increased CLOCK protein expression in CCA cells [21]. Circadian locomotor output cycles protein kaput (CLOCK), a direct target of miR-141 [43], may act as a tumor suppressor by regulating the circadian clock, which can control the cell cycle in response to the DNA damage induced by gemcitabine. These findings suggest that miR-141 modulated gemcitabine cytotoxicity through regulating CLOCK-dependent circadian machinery.

### 2.3. miR-200b

Upregulated miR-200b was found in three malignant cholangiocyte cell lines, compared with nonmalignant cholangiocytes [21]. Repression of miR-200b increased gemcitabine-induced cytotoxicity in malignant cholangiocytes, while the effect was decreased upon ectopic expression of miR-200b, suggesting a role of miR-200b in modulating sensitivity to gemcitabine cytotoxicity. Inhibition of miR-200b increased the expression of protein tyrosine phosphatase, nonreceptor type 12 (PTPN12), leading to the decreased phosphorylation of Src at tyrosine 182, which is essential to mediate downstream signal transduction for cell proliferation and differentiation. Overall, miR-200b can modulate the sensitivity of malignant cholangiocytes to gemcitabine through regulating PTPN12/Src signaling [21].

### 2.4. Let-7a

Let-7a, the first identified miRNA, is a tumor-suppressor miRNA in various cancers such as colorectal cancer [44], hepatocellular carcinoma [45], lung cancer [46], glioma [47], and renal cell carcinoma [48]. Let-7a was found to be downregulated in opisthrorchiasis-associated CCA [49]. Interleukin-6 (IL-6) mediated activation of signal transducers and activators of transcription 3 (STAT3) was aberrantly sustained in CCA through enhancing myeloid cell leukemia 1 (Mcl-1) expression, resulting in the resistance to apoptosis [50]. Let-7a expression was increased in IL-6-overexpressing malignant cholangiocytes, both in vivo and in vitro [22]. Let-7a decreased gemcitabine-induced apoptosis via directly targeting neurofibromatosis 2 (NF2) [22], which is a negative regulator of the phosphorylation of signal transducer, and activator of transcription 3 (STAT3) [51]. These findings indicate a novel mechanism of the Let-7a/NF2 axis, involving IL-6/STAT3 signaling to promote tumor growth and chemoresistance in human malignant cholangiocytes.

### 2.5. miR-29b

The miRNA-29 family includes miR-29a, miR-29b-1/2, and miR-29c. Aberrant expression of these miRNAs is frequent in various cancers, acting as tumor suppressors or oncogenes in a cellular context-dependent manner. Several studies have shown that miR-29b is downregulated and serves as a tumor suppressor in acute myelogenous leukemia [52], lung cancer [53], glioblastoma [54], breast cancer [55], and pancreatic cancer [56]. Downregulated miR-29b was also found in malignant cholangiocytes [57], which could be due to the regulation of the miR-29b promoter region by c-Myc, NF-kB, and the hedgehog signaling pathway [58]. Ectopic expression of miR-29b sensitized KMCH CCA cells to TNF-related apoptosis-inducing ligand (TRAIL)-induced apoptosis through directly suppressing Mcl-1. On the other hand, the silencing of miR-29b in normal cholangiocytes increased Mcl-1 protein expression, resulting in reduced TRAIL-mediated apoptosis. These findings indicated that the miR-29b/Mcl-1 axis contributes to the malignant phenotype.

The effect of miR-29b on regulation of gemcitabine cytotoxicity in CCA has been demonstrated by Okamoto et al. [30]. The expression of miR-29b was lower in HuH28 cells, a cholangiocarcinoma cell line, compared with HuCCT1 cells, another cholangiocarcinoma cell line with greater sensitivity to gemcitabine. Overexpression of miR-29b in HuH28 cells increased gemcitabine-induced cytotoxicity. They found that the transfection of miR-29b mimics the reduced protein expression of PI3K regulatory subunit 1 (PI3KR1) and matrix metalloproteinase 2 (MMP-2), which are predicted targets of miR-29b. Silencing of PI3KR1 or MMP-2 expression further decreased the viability of HuH28 cells upon gemcitabine treatment. These results suggest a role of miR-29b in regulating the gemcitabine response through targeting PI3KR1 or MMP-2.

### 2.6. miR-205

Upregulated miR-205 was found in head and neck cancer and ovarian cancer [59,60], while downregulated miR-205 was frequent in the development of advanced cancers such as breast cancer [61], gastric cancer [62], and bladder cancer [63]. Downregulated miR-205 has been shown to act as a repressor of EMT by targeting ZEB1 and ZEB2 [64]. The expression of miR-205 may be higher in epithelial cell-dominated cancer tissues with a better prognosis, and lower in those invasive cancers with a worse prognosis [65].

In the study by Okamoto et al., lower expression of miR-205 in HuH28 cells was also found. Ectopic expression of miR-205 further reduced viability upon gemcitabine treatment. While erythroblastic leukemia viral oncogene homolog 3 (ErbB3) or vascular endothelial growth factor A (VEGFa) were predicted targets of miR-205, altered expression of ErbB3 and VEGFa was not observed upon overexpression of miR-205, suggesting that other target genes are involved in the regulation of gemcitabine sensitivity by miR-205 in CCA [65].

### 2.7. miR-221

The miR-221 family, composed of miR-221 and miR-222, includes oncogenic miRNAs (oncomirs) with overexpression levels in cancers [66]. The expression level of miR-221 in extrahepatic CCA was significantly higher than in paired para-carcinoma tissues [67]. Higher miR-211 expression was associated with metastasis and poorer prognosis. Ectopic expression of miR-211 promoted EMT of extrahepatic CCA cells through directly suppressing PTEN and forming a positive feedback loop of the β-catenin/cJun signaling 1pathway. On the other hand, inhibition of miR-211 reduced the migration and invasion of malignant cholangiocytes by modulating the PTEN-dependent β-catenin/cJun signaling pathway [67]. These results suggest a role of miR-211 in promoting EMT in extrahepatic CCA. In intrahepatic CCA, however, miR-211 expression was downregulated compared with normal tissues, but not significantly associated with prognosis [68]. Similar to the role of miR-29b in the study by Okamoto et al., overexpression of miR-211 restored gemcitabine-mediated cytotoxicity by targeting PI3KR1 [30].

### 2.8. miR-181c

In cancer development, miR-181c may function differently depending on the different cellular contexts of distinct tumor types. Upregulated miR-181 manifested as an oncogene in Ewing’s sarcoma [69], pancreatic cancer [70], breast cancer [71], and hepatocellular carcinoma [72]. However, miR-181c could serve as a tumor suppressor in gastric cancer [73] and glioblastoma [74]. Wang et al. reported that upregulated miR-181c was found in CCA tissues [23]. Overexpression of miR-181c contributed to cholangiocarcinogenesis and metastasis by targeting N-myc Downstream-Regulated Gene 2 (NDRG2). NDRG2 is known as a tumor suppressor gene through inhibition of tumor growth and metastasis [75]. Moreover, leukemia inhibitory factor (LIF) induced miR-181c expression in human CCA. NDRG2 overexpression abolished mothers against decapentaplegic homolog 2 (Smad2), leading to decreased expression of LIF. These findings establish that LIF/miR-181c and NDRG2 could counteract each other, indicating a novel mechanism for regulating CCA development and metastasis. Furthermore, NDRG2 overexpression enhanced gemcitabine-mediated inhibition of tumor growth, suggesting the role of miR181c/NDRG2 signaling in modulating gemcitabine response in CCA.

### 2.9. miR-130a-3p

Downregulated miR-130a-3p was found in several cancer types, including nasopharyngeal cancer [76], esophageal cancer [77], gastric cancer [78], and triple-negative breast cancer [79], suggesting a role of the tumor suppressor in cancer development. Moreover, miR-130a-3p was found to be involved in regulating the cisplatin response in esophageal squamous cell carcinoma [80] and gastric cancer [81], and gemcitabine resistance in hepatocellular carcinoma [82]. Asukai et al. reported that tumors with higher expression of miR-130a-3p were associated with poorer survival in patients with CCA [25]. The expression level of miR-130a-3p is upregulated in two gemcitabine-resistant CCA cells compared with their parental cells. Ectopic expression of miR-130a-3p in parental CCA cells increased gemcitabine resistance. Peroxisome proliferator-activated receptor-γ (PPARγ) is the direct target of miR-130a-3p. Pioglitazone, a PPARγ agonist, could alleviate gemcitabine resistance in CCA. These results indicated that miR-130-3p contributed to gemcitabine resistance through suppressing PPARγ.

### 2.10. miR-1249

Aberrant expression of miR-1249 has previously been reported in several cancer types. Upregulated miR-1249 was found to promote glioma growth by directly targeting adenomatous polyposis coli 2 [83]. MiR-1249, as a transcriptional target of p53, inhibited tumor growth, metastasis, and angiogenesis by targeting VEGFa and the high mobility group AT-hook 2 (HMGA2) in colorectal cancer [84]. Overexpressed miR-1249-3p enhanced tumor growth and invasion of hepatocellular carcinoma [85].

In intrahepatic CCA tissues, miR-1249 was overexpressed in tumor tissues compared with adjacent normal tissues [24]. Cases with higher miR-1249 were associated with worse prognosis, independently of adjuvant chemotherapy. Inhibition of miR-1249 enhanced the sensitivity of doublet chemotherapy, including cisplatin and gemcitabine. Of note, a cytotoxic effect of miR-1249 inhibitor was not observed in the absence of chemotherapy treatment. Increased expression of miR-1249 was found in subpopulations of CCA cells that expressed CD-13 or CD-133, which are common markers of chemoresistant cancer cells. Furthermore, miR-1249 induced the expansion of CD133+ CCA cells by targeting frizzled class receptor 8 (FZD8). FZD8 could act as a negative regulator of the canonical Wnt pathway by activating the noncanonical Wnt/calcium pathway [86]. Altogether, miR-1249 mediated the expansion of CD133+ CCA cells with chemoresistance, by targeting FZD8/Wnt signaling [24].

### 2.11. miR-210

Hypoxia stress modulates the expression of a subset of miRNAs named hypoxamiRs [87]. Among hypoxamiRs, miR-210 is one of the master hypoxamiRs dominantly induced by hypoxia, exhibiting oncogenic properties in many cancer types, including CCA [88]. In a mouse model of cholestasis-associated CCA, hypoxia-induced-factor-2α (HIF2α) upregulated miR-210 expression, suppressing MAX network transcriptional repressor (Mnt), which antagonizes cMyc by competing binding to MAX [88]. As a result, activation of c-Myc promotes cholangiocarcinogenesis through upregulation of cyclin D1.

Increased miR-210 expression in CCA tissues was shown, compared with normal bile duct tissues, and associated with shorter survival [26]. In a pseudohypoxia status induced by CoCl_2_ treatment, miR-210 suppressed the proliferation of malignant cholangiocytes through increasing cell cycle arrest at G2/M phase [26]. Furthermore, miR-210 decreased gemcitabine sensitivity by targeting HIF-3α, a negative regulator of HIF-1α. Overall, Silakit et al. demonstrated that the positive feedback loop of miR-210/HIF-1α, by regulating HIF-3α, may contribute to cell arrest and gemcitabine resistance of CCA under hypoxia [26].

## 3. MiRNAs and Resistance to Fluorouracil or Cisplatin

5-fluorouracil (5-FU), a synthetic pyrimidine analogue of uracil, inhibits thymidylate synthase, restricting the availability of thymidine nucleotides for RNA and DNA synthesis. Similar to gemcitabine, 5-FU belongs to the antimetabolite agents. On the other hand, cisplatin forms monoadducts or interstrand or intrastrand crosslinks between purine bases and inhibits repair of DNA damage. Either gemcitabine or 5-FU, combined with cisplatin, are commonly used as active treatments for CCA. However, the limited duration of CCA cells’ response to 5-FU or cisplatin warrants further research. We summarize the miRNAs involved in regulating the response of 5-FU or cisplatin in Figure 2.

### 3.1. miR-199a-3p

Reduced miR-199a-3p expression was found in several cancer types, including head and neck squamous cell carcinoma [89], clear cell renal cell carcinoma [90], osteosarcoma [91], and ovarian cancer [92]. Several reports have reported that miR-199a-3p acts as a tumor suppressor by targeting mTOR in endometrial cancer [93], papillary thyroid carcinoma [94], and hepatocellular carcinoma [95,96].

Downregulated miR-199a-3p was found in malignant cholangiocytes, when compared with normal biliary epithelial cells [97]. Ectopic overexpression of miR-199a-3p in cholangiocarcinoma cells increased sensitivity to cisplatin by targeting the mTOR signaling pathway [31]. Moreover, upregulation of miR-199a-3p suppressed cisplatin-induced expression of multidrug resistance protein 1 (MDR1), suggesting that miR-199a-3p enhanced cisplatin cytotoxicity by regulating MDR1 [31]. These findings indicated that miR-199a-3p could increase cisplatin sensitivity to CCA through regulating mTOR and MDR1.

### 3.2. miR-200b/c

The miR-200 family (miR-141, miR-200a,b,c, and miR-429) is one of the first groups of miRNAs shown to regulate EMT [64]. These miRNAs exhibit tumor suppressor activity to antagonize EMT associated with cancer metastasis by silencing E-cadherin, such as ZEB1 and ZEB2 in various cancers such as colorectal cancer [98], gastric cancer [99], pancreatic cancer [100], ovarian cancer [101], and nasopharyngeal cancer [102].

The expression of miR-200b/c was downregulated in CCA tissues compared with normal bile duct tissues [32]. Upregulation of miR-200b/c inhibited migration, invasion, and distant metastasis of CCA cells, while silencing miR-200b/c promoted the metastatic process through modulating Zeb1/2 and the Rho-associated protein kinase 2 (ROCK2) pathway. ROCK2 has been shown to promote actin‒myosin contractility and tubulin polymerization by activating LIM kinase 2 (LIMK2), which abolishes the ability of cofilin to depolymerize actin [103]. On the other hand, miR-200b/c could regulate tumor initiation or sphere formation of CCA by targeting suppressor of zeste 12 homolog (SUZ12), which is one cofactor essential for enhancer of zeste homolog 2 (EZH2) to catalyze trimethylation of histone at lysine 27 (H3K27me3), leading to transcriptional repression of target genes. Moreover, overexpression of miR-200b/c enhanced 5-FU sensitivity to CCA cells, while the direct target was not reported in this study [32].

### 3.3. miR-106b

The miR-106b-25 cluster is composed of highly conserved miR-106b, miR-93, and miR-25. MiR-106b may have varied functions, and acts as an oncogene in different types of cancers. For example, miR-106b promoted tumor cell proliferation, migration, and invasion in breast cancer [104], gastric cancer [105], and hepatocellular carcinoma [106].

Zinc finger and BTB domain containing 7A (ZBTB7A) is a transcriptional repressor of the POZ/BTB and Krüppel (POK) family transcription factors, acting as a proto-oncoprotein or a tumor suppressor, depending on the cellular context [107]. For example, overexpression of miR-106b inhibited apoptosis by directly targeting ZBTB7A in hepatocellular carcinoma [108]. In contrast, miR-106b exhibited the tumor-suppressive function through regulating ZBTB7A in ovarian cancer [109].

The expression of miR-106b was downregulated in CCA tumors [33]. In 5-FU-resistant CCA cells, miR-106b expression was downregulated compared with parental CCA cells. Ectopic expression of miR-106b resensitized 5-FU-resistant CCA cells to 5-FU by targeting ZBTB7A, suggesting that miR-106b/ZBTB7A axis could modulate 5-FU resistance in CCA.

### 3.4. miR-320

Several studies identified that miR-320 family members were downregulated, and inhibited tumor cell proliferation, metastasis, or angiogenesis in colon cancer [110], tongue squamous cell carcinoma [111], cervical cancer [112], breast cancer [113], prostate cancer [114], and glioma [115]. These results indicated that miR-320 functions as a tumor suppressor in cancer progression.

MiR-320 was downregulated in intrahepatic CCA [17]. Chen et al. reported that downregulation of miR-320 restored the expression of Mcl-1, an antiapoptotic Bcl-2 family member, thus avoiding mitochondrial-dependent cell apoptosis [17]. Furthermore, overexpression of miR-320 in malignant cholangiocytes promoted 5-FU-induced apoptosis. Taken together, miR-320 could increase sensitivity to 5-FU by targeting Mcl-1 in CCA.

## 4. MiRNAs and Resistance to Targeted Drugs

A number of emerging targeted drugs have been investigated to treat CCA, such as FGFR2 small molecule kinase inhibitors, mutant IDH inhibitors, Mcl-1 selective inhibitors, and MEK inhibitors [116]. Previously, several tyrosine kinase inhibitors in the treatment of CCA have been reported, but their efficacy is not promising. Sorafenib is a tyrosine kinase inhibitor directly inhibiting several targets, including VEGF receptor (VERFR) 2/3, platelet-derived growth factor receptor, Fms-related tyrosine kinase-3, b-RAF, and c-KIT. One phase 2 study revealed no confirmed responsive cases, with 39% stable disease in patients with advanced biliary tract cancer receiving sorafenib alone as the first-line treatment [117]. On the other hand, heat shock protein (Hsp) 90 has been shown to serve as a chaperone of FGFR [118], providing a biochemical basis for targeting CCA with FGFR aberrations. Several miRNAs have been shown to be involved in sorafenib resistance and Hsp90 inhibitors (Figure 3).

### 4.1. miR-138

MiR-138 has been found to function as a tumor suppressor in many cancers, such as cervical cancer [119], esophageal squamous cell carcinoma [120], clear cell renal cell carcinoma [121], and non-small cell lung cancer [122]. In CCA tissues, miR-138 expression was reduced and associated with poor prognosis [34]. Zheng et al. reported that overexpression of miR-138 promoted sensitivity to sorafenib. SRY-related HMG-box 4 (SOX4) was identified as a target of miR-138. SOX4 is a transcriptional factor involved in multiple developmental pathways, including PI3K and Wnt signaling, and serving as an oncogene in many cancers [123]. SOX4 overexpression reversed sorafenib cytotoxicity in malignant cholangiocytes. These findings suggested that miR-138 modulated response to sorafenib through targeting SOX4.

### 4.2. miR-141, miR-330

The organic cation transporter (hOCT1) has been found to regulate the efficacy of sorafenib in CCA [124]. Downregulated hOCT1 was found in CCA. Lozano et al. further investigated the mechanisms resulting in impaired hOCT1 expression, leading to a lack of response to sorafenib [27]. They found that the ectopic expression of miR-141 and miR-330 reduced hOCT1 expression in vitro. Furthermore, both miR-141 and miR-330 expression were consistently overexpressed in CCA, and hOCT1 was downregulated in both intrahepatic CCA and extrahepatic CCA. Additionally, they found that the hypermethylated promoter region and aberrant splicing of hOCT1 lead to decreased hOCT1 expression and sorafenib response. Ectopic expression of hOCT1 increased sorafenib uptake in malignant cholangiocytes, enhancing the cytotoxic effect of sorafenib. Of note, it is not clear from the study if hOCT1 is the predicted target of miR-141 and/or miR-330, or if other targets are involved in regulating hOCT1 [27].

### 4.3. miR-21

Lampis et al. identified that Hsp 90 inhibitors are effective in CCA cell lines from a high-throughput screening [28]. They further found that higher miR-21 expression was associated with a 50% higher growth inhibitory concentration (GI50) of AUY922, a Hsp90 inhibitor. Inhibition of miR-21 restored sensitivity to AUY922 in CCA cell lines but enforced expression of miR-21 increased resistance, suggesting the role of miR-21 in driving resistance to Hsp90 inhibitors. DnaJ heat shock protein family member B5 (DNAJB5) was identified as the direct target of miR-21. DNAJB5 is a member of the DNAJ heat shock protein 40 family of co-chaperone proteins, involved in nuclear localization of Histone deacetylase (HDAC). Ectopic expression of DNAJB5 in miR-21 overexpressing cells resensitized CCA cells to AUY922, suggesting that DNAJB5 mediated miR-21-dependent resistance to AUY922. Overall, these results indicated that miR-21 mediated resistance of Hsp90 inhibitors by targeting DNAJB5 in CCA [28].

### 4.4. miR-25

Along with miR-93 and miR-106b, miR-25 is a member of the miR-106b-25 cluster, which is dysregulated in many cancers. Several studies reported that miR-25 plays an oncogenic role in esophageal cancer [125], gastric cancer [126], and lung cancer [127]. On the other hand, miR-25 was found to have a tumor-suppressor function in thyroid cancer and colon cancer [128,129]. Increased miR-25 expression was found in CCA cell lines and tumor samples [29]. Of note, increased RNA expression of minichromosome maintenance 7 (MCM7) was also found in malignant cholangiocytes, and tumor samples with high miR-25 expression. Given that miR-25 is located within the intron 13 of MCM7 gene locus, these findings may explain in part the elevated levels of miR-25. Enforced expression of miR-25 protected CCA cells from TNF-related apoptosis-inducing ligand (TRAIL)-induced apoptosis. Antagonism of miR-25 increased death receptor 4 (DR4) protein expression, sensitizing malignant cholangiocytes to apoptosis. DR4 is one of the TRAIL death receptors’ direct targets of miR-25. Enforced DR4 expression also restored sensitivity to TRAIL-induced apoptosis. These results indicated that miR-25 regulates TRAIL-mediated apoptosis of CCA by targeting DR4 [29].

## 5. MiRNAs as Therapeutic Targets in Cholangiocarcinoma

### 5.1. miRNA-Based Therapeutics in Cancer

Owing to their essential role in tumor pathogenesis and progression, miRNAs have been proposed as novel therapeutic targets [130]. There are two miRNA-based therapeutic approaches: miRNA replacement and miRNA inhibition. The former approach introduces synthetic miRNA mimics or viral vectors expressing RNA to restore downregulated miRNA function. For the miRNA inhibition approach, aberrantly overexpressed miRNAs are suppressed using anti-miR oligonucleotides (AMOs), locked nucleic acid (LNA) anti-miRNAs, or miRNA sponges, such as circular RNA and long non-coding RNA. However, miRNA-based therapeutics pose several challenges, including delivery to specific target cells, avoiding degradation, and reducing toxicity. For example, naked miRNAs are easily degraded in the bloodstream by ribonuclease and do not easily pass through the extracellular matrix. Inherent properties, such as hydrophilicity and negative charge, hinder penetration across cell membranes. After endocytosis, endosome escape and resistance to cytoplasmic ribonuclease are critical for synthetic miRNAs to exert their therapeutic effect [131,132]. Systemic administration of miRNAs, like other single-stranded or double-stranded RNAs, may trigger immunotoxicity with secretion of type I interferon through Toll-like receptors [133]. While chemical modifications, such as phosphorothioate containing oligonucleotides, or locked nucleic acid oligonucleotides, increased the stability of synthetic miRNAs, the loss of efficiency of loading into the RNA-induced silencing complex limits the development of clinical applications. Viral and nonviral vectors have been developed to improve the efficiency of delivery into target cells. While viral vectors, such as adenoviruses and lentiviruses, have advantages, including efficient transfection and constant expression of genes, the risk of adverse immunogenicity, and viruses in production, restrict the use of viral systems. On the other hand, nonviral vectors, including lipid-based and polymer-based carriers, have been more promising for the further development of gene therapy, due to safety and the low immune response. Several miRNA-based therapeutics are in early clinical trials according to www.clinicaltrials.gov. A recent study by Hong et al. revealed the first clinical experiment with miRNA-based treatment in oncology [134]. Eighty-five patients with advanced solid tumors were intravenously given a daily dose of liposomal mimic of miR-34a (MRX34) in three-week cycles. Three patients had partial responses and 16 had durable, stable disease. The study was halted early due to serious immune-related adverse events, resulting in four deaths. These results raised concerns about the safety of this new class of drug, requiring the further development of dosing and premedication use to avoid immune-related toxicity. On the other hand, while the pharmacodynamics results confirmed a proof-of-concept for miRNA-based treatment, whether the antitumor activity or toxicity of MRX34 is related to specific gene-suppressing function or other immune-related mechanisms requires further exploration.

### 5.2. The Potential Role of miRNA-Based Therapeutics in Cholangiocarcinoma

Li et al. demonstrated bidirectional communication between tumor and liver stromal cells through miRNA-loaded extracellular vesicles, providing the basis for delivering miRNAs as a novel approach to increase antitumor effects [135]. They first found that CCA cells downregulated the expression of miR-195 in co-cultured stromal cells, while miR-195-overexpressing stromal cells inhibited CCA cells’ growth, invasion, and migration. Extracellular vesicles loaded with miR-195 could inhibit CCA growth in vivo and in vitro. Of note, they showed that extracellular vesicles derived from stromal cells selectively accumulated in CCA but not in normal liver parenchyma in vivo [135]. Enhanced permeability and retention effects might contribute to this phenotype [136]. However, further studies on the specific tropism for these extracellular vesicles toward CCA cells are warranted. The study by Li et al. contributed to the current knowledge of intercellular communications between CCA and microenvironment, suggesting a potential therapeutic role of miR-195 mimics in CCA. Xie et al. developed cholesterol-modified polymeric C-X-C receptor type 4 (CXCR4) antagonist (PCX) nanoparticles, with codelivery of anti-miR-210 to cooperatively exert antitumor activity in CCA [137]. Inhibition of CXCR4 signaling decreases CCA cells migration and invasion [138,139]. As mentioned in Section 2.11, miR-210, which is induced by hypoxia, exerts oncogenic activity to promote CCA growth. PCX/anti-miR-210 nanoparticles induced apoptosis of CCA cells and sensitized CCA cells to combination chemotherapy with gemcitabine/cisplatin. The antitumor effect was also demonstrated in a CCA xenograft model. The results showed that an innovative nanotherapeutic approach that combined inhibition of CXCR4 and miR-210 could efficiently kill CCA cells through the induction of apoptosis [137].

As gemcitabine is an active chemotherapy agent for CCA, a combination of gemcitabine and miRNAs to create a synergistic antitumor effect could be a potential therapeutic approach. However, the efficacy and biosafety for systemic administration remain major concerns that may delay the further clinical application. Zhang et al. assembled amphiphilic gemcitabine‒oleic acid prodrugs (GOA) binding miRNAs with hydrogen bonds into nanoparticles to form GOA/miR nanoparticles [140]. They demonstrated that GOA/miR-122 nanoparticles accumulated significantly in tumors and inhibited tumor growth in vivo, without inducing significant inflammation or the alteration of biochemical indicators, indicating their efficacy and biosafety using negatively charged GOA/miR nanoparticles. However, owing to the cancer model in this study being hepatocellular carcinoma [140], the function and biosafety of the noncationic GOA/miR nanoparticles in CCA warrant further research.

## 6. Conclusions

An increasing number of miRNAs have been identified as involved in chemoresistance or targeted drug resistance in CCA. As summarized in the present review, several miRNAs play a significant role in modulating drug response to gemcitabine, cisplatin, and 5-FU, by targeting specific genes that have been shown to exhibit an oncogenic or tumor-suppressive function in other cancer types. A variety of studies demonstrated that counterexpression of miRNAs in resistant cells could resensitize these cells to chemotherapeutic or targeted agents, supporting the potential of miRNAs as an adjunct to traditional anticancer treatment, to enhance cancer cell killing activity. However, it is still a challenge to implement miRNA-based therapeutics in patients. As several targeted therapies might change the treatment paradigm in at least a minority of patients with CCA [116], further studies will continue to shed light on potential miRNA targets involved in targeted drug resistance. To promote miRNA-based therapeutics that surmount drug resistance or enhance antitumor effects in CCA, future studies should focus on nanomedicine to facilitate the synergistic delivery of chemotherapy and miRNAs.

## Figures and Tables

**Figure 1 biomolecules-10-01396-f001:**
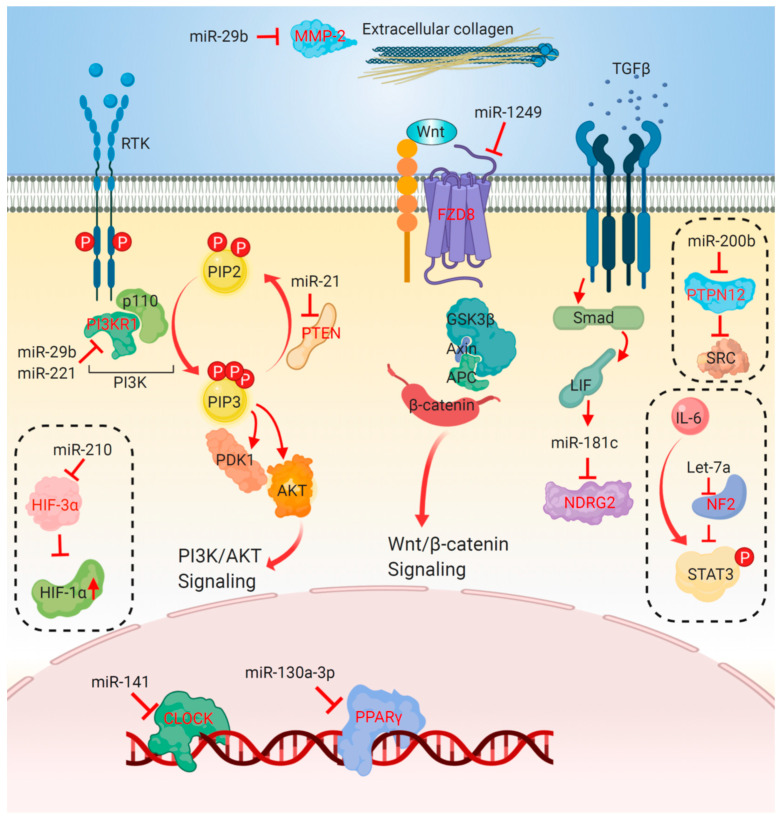
Mechanisms of microRNA (miRNA)-mediated modulating response to gemcitabine. Overexpression of miR-21 reduced gemcitabine cytotoxicity through enhancing PI3K/Akt pathway via inhibiting PTEN. Upregulation of miR-141 and miR200b increased resistance to gemcitabine via targeting CLOCK and PTPN12. Let-7a directly suppressed NF2, leading to upregulated phosphorylation of STAT3 and gemcitabine resistance. PI3KR1 and MMP2 were targets of miR-29b, which enhanced gemcitabine cytotoxicity. Similarly, miR-211 restored gemcitabine sensitivity by inhibiting PI3KR1. Overexpression of miR-181c increased gemcitabine resistance by targeting NDRG2. Mir-130-3p resulted in gemcitabine resistance via suppressing PPARγ. Upregulation of miR-1249 increased chemoresistance by targeting FZD8, a negative regulator of the canonical Wnt pathway. Under hypoxia condition, miR-210 contributed to cell arrest of malignant cholangiocytes by regulating HIF3α, leading to gemcitabine resistance. The arrow at line end indicates activation, while the bar at line end indicates inhibition. Direct targets of miRNAs are depicted in red lettering. Abbreviations: PTEN, phosphatase and tensin homolog; PI3K, phosphoinositide 3-kinase; PI3KR1, PI3K regulatory subunit 1; Akt, protein kinase B; PDK1, phosphoinositide-dependent kinase-1; CLOCK, circadian locomotor output cycles protein kaput; PTPN12, protein tyrosine phosphatase non-receptor type 12; NF2, neurofibromatosis 2; STAT3, signal transducer and activator of transcription 3; Smad2, mothers against decapentaplegic homolog 2; NDRG2, N-myc downstream-regulated gene 2; MMP-2, matrix metalloproteinase 2; PPARγ, peroxisome proliferator-activated receptor-γ; HIF-3α, hypoxia-induced-factor-3α; HIF-1α, hypoxia-induced-factor-1α.

**Figure 2 biomolecules-10-01396-f002:**
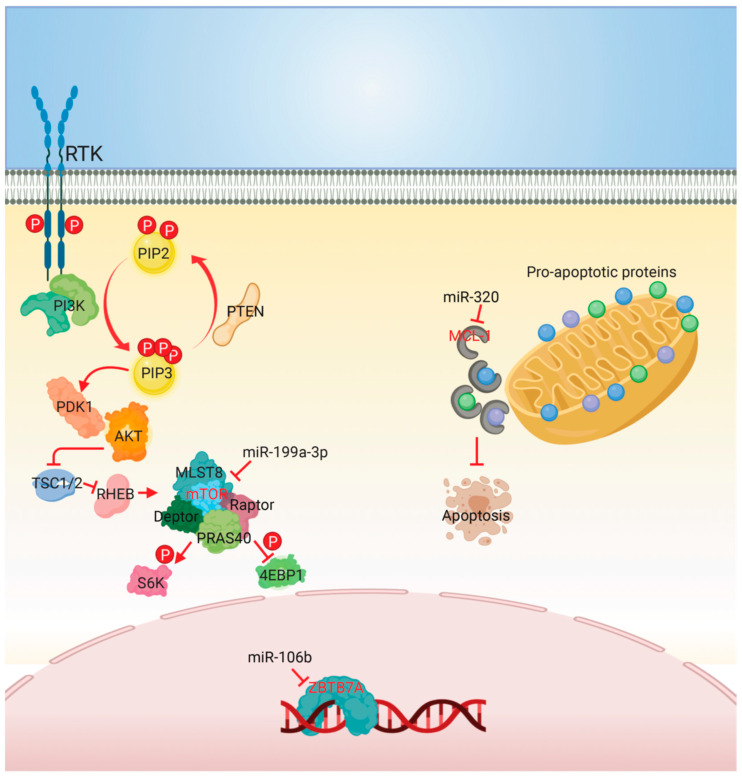
Mechanisms of miRNAs involved in modulating sensitivity to 5-fluorouracil and cisplatin. The mTORC1 complex, consisting of mTOR, raptor, deptor, mLST8, and PRAS40, activates S6K and inhibits 4EBP1 through phosphorylation. Overexpression of miR-199a-3p upregulated cisplatin sensitivity by directly targeting mTOR. Overexpression of miR-106b resensitized 5-FU resistant CCA cells by targeting ZBTB7A. Downregulated miR-320 increased Mcl-1 expression, resulting in reducing 5-FU-induced apoptosis. The arrow at line end indicates activation, while the bar at line end indicates inhibition. Direct targets of miRNAs are depicted in red lettering. Abbreviations: PTEN, phosphatase and tensin homolog; PI3K, phosphoinositide 3-kinase; PI3KR1, PI3K regulatory subunit 1; Akt, protein kinase B; PDK1, phosphoinositide-dependent kinase-1; mTOR, the mammalian target of rapamycin; ZBTB7A, zinc finger and BTB domain containing 7A; Mcl-1, myeloid cell leukemia 1.

**Figure 3 biomolecules-10-01396-f003:**
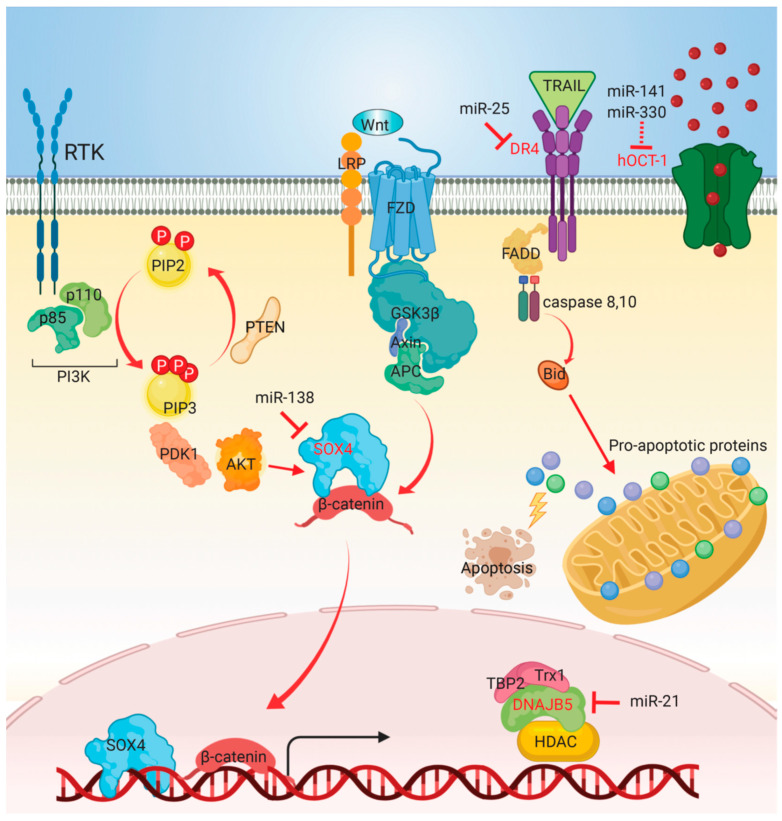
Mechanisms of miRNAs involved in modulating sensitivity to targeted therapies. TRAIL/DR4 signaling induces apoptosis through recruiting FADD and caspase 8 or 10 to form the death-receptor complex. Trx1 forms a stable complex with TBP-2 and DNAJB5, reducing HDAC4 activity. Upregulation of miR-138 promoted sorafenib sensitivity by targeting SOX4, which is a transcriptional factor involved in oncogenic signaling pathways. Overexpression of miR-141 and miR-330 reduced hOCT1 expression and therefore decreased sorafenib uptake. MiR-21 mediated resistance of Hsp90 inhibitors by targeting DNAJB5. Inhibition of miR-25 increased DR4 expression, sensitizing malignant cholangiocytes to TRAIL-induced apoptosis. The arrow at line end indicates activation, while the bar at line end indicates inhibition. Direct targets of miRNAs are depicted in red lettering. Abbreviations: PTEN, phosphatase and tensin homolog; PI3K, phosphoinositide 3-kinase; PI3KR1, PI3K regulatory subunit 1; Akt, protein kinase B; PDK1, phosphoinositide-dependent kinase-1; SOX4, SRY-related HMG-box 4; hOCT1, organic cation transporter 1; DNAJB5, DnaJ heat shock protein family member B5; Trx1, thioredoxin1; TBP-2, thioredoxin binding protein-2; HDAC, histone deacetylase; DR4, death receptor 4; Hsp, heat shock protein; TRAIL, TNF-related apoptosis-inducing ligand; FADD, Fas-associated protein with death domain.

**Table 1 biomolecules-10-01396-t001:** Summary of the upregulation/downregulation of miRNAs and their targets involved in the drug resistance of cholangiocarcinoma.

miRNA	Expression	Target	Signaling	Resistance	Reference
miR-21	Up	PTEN	PI3K/Akt	Gemcitabine	[21]
miR-141	Up	CLOCK	-	Gemcitabine	[21]
miR-200b	Up	PTPN12	Src	Gemcitabine	[21]
Let-7a	Up	NF2	STAT3	Gemcitabine	[22]
miR-181c	Up	NDRG2	-	Gemcitabine	[23]
miR-1249	Up	FZD8	Wnt/β-catenin	Gemcitabine	[24]
miR-130a-3p	Up	PPARγ	-	Gemcitabine	[25]
miR-210	Up	HIF-3α	-	Gemcitabine	[26]
miR-141, miR-330	Up	hOCT1	-	Sorafenib	[27]
miR-21	Up	DNAJB5	-	Hsp90 inhibitor	[28]
miR-25	Up	DR4	caspase 8/3	TRAIL	[29]
miR-29b	Down	PI3KR1	PI3K/Akt	Gemcitabine	[30]
miR-29b	Down	MMP-2	-	Gemcitabine	[30]
miR-205	Down	-	-	Gemcitabine	[30]
miR-221	Down	PI3KR1	-	Gemcitabine	[30]
miR-199a-3p	Down	MDR1, mTOR	mTOR signaling	Cisplatin	[31]
miR-200b/c	Down	-	-	5-FU	[32]
miR-106b	Down	ZBTB7A	-	5-FU	[33]
miR-320	Down	Mcl-1	-	5-FU	[17]
miR-138	Down	SOX4	-	Sorafenib	[34]

Abbreviations: PTEN, phosphatase and tensin homolog; PI3K, phosphoinositide 3-kinase; Akt, protein kinase B; CLOCK, circadian locomotor output cycles protein kaput; PTPN12, protein tyrosine phosphatase nonreceptor type 12; NF2, neurofibromatosis 2; STAT3, signal transducer and activator of transcription 3; NDRG2, N-myc downstream-regulated gene 2; PI3KR1, PI3K regulatory subunit 1; MMP-2, matrix metalloproteinase 2; PPARγ, peroxisome proliferator-activated receptor- γ; HIF-3α, hypoxia-induced-factor-3α; MDR1, multidrug resistance protein 1; mTOR, the mammalian target of rapamycin; ZBTB7A, zinc finger and BTB domain containing 7A; Mcl-1, myeloid cell leukemia 1; SOX4, SRY-related HMG-box 4; hOCT1, organic cation transporter 1; DNAJB5, DnaJ heat shock protein family member B5; DR4, death receptor 4; Hsp, heat shock protein; TRAIL, TNF-related apoptosis-inducing ligand.

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
