# Peer review of "The Emerging Role of MicroRNAs in Regulating the Drug Response of Cholangiocarcinoma"

_biomolecules, 2020, doi:10.3390/biom10101396_

Round 1
Reviewer 1 Report
The review “The emerging role of microRNAs in regulating the drug response of cholangiocarcinoma”, by Huang et al. summarizes the current knowledge on microRNAs linked to drug resistance in cholangiocarcinoma (CCA). The manuscript describes in detail the MiRNAs, the up/down regulation, targets and signaling involved in the CCA resistance to gemcitabine, cisplatin and 5-Fu, which are the current standard first-line treatment for patients with advanced CCA, as well as targeted therapies.
Authors have carried out a systematic and well-organised bibliographic search. It includes the majority of the articles published in the subject of interest and all references are appropriate and adequate. On the other hand, the figures are very clear and well-designed.
However, in my opinion, several major points should be noted before approval:
- It has recently been published a very similar review on the same type of cancer, describing the role of noncoding RNAs that include microRNAs (miRNAs), circular RNAs (circRNAs) and long noncoding RNAs (lncRNAs) in cholangiocarcinoma, covering more general aspects (Lv Y, Wang Z, Zhao K, Zhang G, Huang S, Zhao Y. Role of noncoding RNAs in cholangiocarcinoma (Review). Int J Oncol. 2020;57(1):7-20. doi:10.3892/ijo.2020.5047). The originality of the present article is moderate, but lies in a more specific approach, pairing miRNA with the mechanisms of resistence to treatment with gemcitabine cisplatin, 5-Fu and targeted therapies.
Please, state clearly in the introduction the importance and the contribution to the field.
- The manuscript is comprehensively and well-organized in sections: “MiRNAs and resitance to gemcitabine, MiRNAs and resitance to fluorouracil or cisplatin and MiRNAs and resistance to targeted drugs”. In my opinion, the authors could use other classification within these sections related to the “target sites” of miRNA, for example, extracellular, citoplasmatic and nuclear locations. It will make the manuscript easier to read instead to list one by one the diferent MiRNAs.
- In the introduction section, lines 43-45, authors affirm that “ the molecular profiling suggests that grouping together CCA subtypes in a clinical trial rather than stratifying patients based on molecular pathogenesis may lead to misinterpretation of treatment efficacy”. However, being this limitation so important, throughout the text any consideration is done related to CCA subtypes linked to MiRNAs and drug resistance.
- miR195 seems important in cholangiocarcinoma miRNA-based therapy according to Li et al., (section 5.2). There is no information described in the text about this particular MiRNA.
- Authors point out in line 457 that “...Further studies on the specific tropism for these extracellular vesicles toward CCA cells are warrented”. This selective high local concentration might be due to the enhanced permeability and retention (EPR) effect, that is based on tumor blood flow and the nano-sized vehicles (see Shi Y, van der Meel R, Chen X, Lammers T. The EPR effect and beyond: Strategies to improve tumor targeting and cancer nanomedicine treatment efficacy. Theranostics. 2020;10(17):7921-7924).
Minor points:
- Lines 61 and 64: correct. It should be “CCA” instead “CGA”
- Lines 64 and 65: The text is not clear. The phrase is confusing, maybe some words are deleted.
- Table 1 legend. Correct “up/downregulation”
- Table 1: in text, complete the blank spaces with a “straight” or with “not known” to avoid missunderstanding
- Figure 3. Add the respective targeted therapy close to their involved miRNA inside the figure for better clarity and understanding (optional)
- Keywords: add “chemotherapy” and “cisplatin”
Author Response
We thank the reviewer for helpful comments and suggestions. Each issue has been addressed and revisions have been made in the manuscript accordingly. All changes are marked by track changes. Our specific answers to the reviewer’s comments are detailed below.
[Major comments]
Point 1: It has recently been published a very similar review on the same type of cancer, describing the role of noncoding RNAs that include microRNAs (miRNAs), circular RNAs (circRNAs) and long noncoding RNAs (lncRNAs) in cholangiocarcinoma, covering more general aspects (Lv Y, Wang Z, Zhao K, Zhang G, Huang S, Zhao Y. Role of noncoding RNAs in cholangiocarcinoma (Review). Int J Oncol. 2020;57(1):7-20. doi:10.3892/ijo.2020.5047). The originality of the present article is moderate, but lies in a more specific approach, pairing miRNA with the mechanisms of resistence to treatment with gemcitabine cisplatin, 5-Fu and targeted therapies.
Please, state clearly in the introduction the importance and the contribution to the field.
Response 1: Thanks to the reviewer for this comments. We checked the review paper mentioned and found the distinct direction from ours. They summarized the role of non-coding RNA including miRNA, circular RNA, and lncRNA, in tumorigenesis, EMT, multidrug resistance, migration, and invasion of CCA. We are more specifically focusing on only miRNAs involved in chemotherapy or targeted therapy response. To emphasize our aim in this review, we modified this part as the reviewer’s suggestion. Please see page 2, line 91-94.
Before |
After |
In the following paragraphs, we summarize the literature on miRNAs linked to drug resistance in CCA. |
Moreover, combined miRNA-based therapeutics to improve the drug resistance is a promising treatment strategy. Thus, in the following paragraphs, we summarize the literature on miRNAs linked to drug resistance in CCA and the recent advances in miRNA-based therapeutics in CCA. |
Point 2: The manuscript is comprehensively and well-organized in sections: “MiRNAs and resitance to gemcitabine, MiRNAs and resitance to fluorouracil or cisplatin and MiRNAs and resistance to targeted drugs”. In my opinion, the authors could use other classification within these sections related to the “target sites” of miRNA, for example, extracellular, citoplasmatic and nuclear locations. It will make the manuscript easier to read instead to list one by one the diferent MiRNAs.
Response 2: Thanks to the reviewer’s suggestion. We tend to keep current format and clearly illustrate the target sites using the figures to assist the readers. We also added the mechanisms briefly in each figure legend to improve the understanding.
Point 3: In the introduction section, lines 43-45, authors affirm that “ the molecular profiling suggests that grouping together CCA subtypes in a clinical trial rather than stratifying patients based on molecular pathogenesis may lead to misinterpretation of treatment efficacy”. However, being this limitation so important, throughout the text any consideration is done related to CCA subtypes linked to MiRNAs and drug resistance.
Response 3: Thanks to the reviewer’s comments. We found that the majority of miRNA studies involved in drug resistance did not take the distinct molecular profiling into consideration. Probably it is because traditional chemotherapy like cisplatin, 5-FU or gemcitabine is applied regardless of specific molecular subtypes. But in the future we can expect that studies investigating resistance to FGFR inhibitor or IDH inhibitor will take the molecular subtypes into account.
Point 4: miR195 seems important in cholangiocarcinoma miRNA-based therapy according to Li et al., (section 5.2). There is no information described in the text about this particular MiRNA.
Response 4: As the reviewer mentioned, we rechecked the literature to figure out whether miR-195 was studied for drug response in CCA. However, we did not find any results showing the role of miR-195 in regulating response to chemotherapy such as gemcitabine and targeted drugs in CCA. As a result, we did not add miR-195 in any category in the text.
Point 5: Authors point out in line 457 that “...Further studies on the specific tropism for these extracellular vesicles toward CCA cells are warranted”. This selective high local concentration might be due to the enhanced permeability and retention (EPR) effect, that is based on tumor blood flow and the nano-sized vehicles (see Shi Y, van der Meel R, Chen X, Lammers T. The EPR effect and beyond: Strategies to improve tumor targeting and cancer nanomedicine treatment efficacy. Theranostics. 2020;10(17):7921-7924).
Response 5: Thanks to the reviewer for the comments. We added this opinion in the text.
Please see page 14, line 491-492.
Before |
After |
Further studies on the specific tropism for these extracellular vesicles toward CCA cells are warranted. |
Enhanced permeability and retention effect might contribute to this phenotype [136]. However, further studies on the specific tropism for these extracellular vesicles toward CCA cells are warranted. |
[Minor comments]
Point 1: Lines 61 and 64: correct. It should be “CCA” instead “CGA”
Response 1: We corrected these typo error. Please see page 2, line 72, 74.
Point 2: Lines 64 and 65: The text is not clear. The phrase is confusing, maybe some words are deleted.
Response 2: Due to the minor rearrangement of text by the editorial, we are not sure. We would like to ask for the specific sentences which are not clear.
Point 3: Table 1 legend. Correct “up/downregulation”
Response 3: We changed to the full term upregulation/downregulation for the correction.
Point 4: Table 1: in text, complete the blank spaces with a “straight” or with “not known” to avoid misunderstanding
Response 4: We added the straight symbols in the blank spaces.
Point 5: Figure 3. Add the respective targeted therapy close to their involved miRNA inside the figure for better clarity and understanding (optional)
Response 5: We added the details of mechanism in the figure legend for better understanding.
Point 6: Keywords: add “chemotherapy” and “cisplatin”
Response 6: Thanks to the reviewer for the suggestion. We added the two words in the keywords.
Reviewer 2 Report
This is a quite well-written and comprehensive review of the topic. It does not seem to require any revision.
Author Response
We thank the reviewer for helpful comments and suggestions. Each issue has been addressed and revisions have been made in the manuscript accordingly. All changes are marked by track changes. Our specific answers to the reviewer’s comments are detailed below.
[Major comments]
Point 1: This is a quite well-written and comprehensive review of the topic. It does not seem to require any revision.
Response 1: Thanks to the reviewer for their comments.
Reviewer 3 Report
We reviewed with a great interest the manuscript biomolecules-936813 by Huang and colleagues. In this review, the authors explored the potential of miRNAs in regulating therapies response in CCA. Notably they evaluated the role endorsed by miRNAs in the response to gold standard CCA treatments (i.e. gemcitabine, 5-FU, cisplatin). A exhaustive list of the oncogenic and tumor suppressor miRNAs described for participating to drug response in CCA is developed in this manuscript. Furthermore, innovative miRNA-based approaches with a great potential for cancer treatment are displayed.
Overall, this study is well referenced and address most of the miRNAs regulating the drug response of CCA. The topic is rather novel in the field and pave the way for innovative therapeutic approaches in CCA. I would recommend this work for publication although the comments below may help to improve the quality of this review.
Major comments:
1- In the introductory section, a few words could be added exhibiting the specificity of CCA microenvironment. Indeed, CCA are associated with a highly desmoplastic stroma which could explain the poor drug response in CCA.
2- As mentioned by the authors, CCA display a molecular heterogeneity discriminating iCCA from eCCA. Notably, these disparities are noticeable in terms of molecular signature, patients survival and response to treatments. Therefore, it should be better specified when possible whether the CCA are intrahepatic or extrahepatic.
3- The authors described mainly results from in vitro studies. In order to complete this review and to go a bit further, the authors could propose CCA clinical trials on adjuvant therapies that could be used in the future to support these experimental results (e.g. the french study (PRODIGE 12—ACCORD 18).
4- In section 5, the part on miRNAs sponges could be developed. For instance, the type of molecules able to sponge miRNAs could be mentioned (e.g. lncRNAs, circRNAs). Moreover, some of them are already described for their participation in CCA carcinogenesis through miRNA sponging and could therefore be cited here.
5- A part could be added on miRNA mediated through extracellular vesicles (EVs) as companion biomarker. Indeed, miRNAs contained in EVs or circulating freely are found in liquid biopsies and have a predictive value for response to therapies.
Minor comments:
I also have a few minor writing suggestions:
- line 14: “improved” should be “improve”
- line 30-31: according to the current dogma, hilar CCA are included in the extrahepatic bile ducts category.
- line 34: “carries” should be “carry”
- line 99: “miRNA” should be “miRNAs”
- line 165: “oncogene” should be “oncogenes”
Author Response
We thank the reviewer for helpful comments and suggestions. Each issue has been addressed and revisions have been made in the manuscript accordingly. All changes are marked by track changes. Our specific answers to the reviewer’s comments are detailed below.
[Major comments]
We reviewed with a great interest the manuscript biomolecules-936813 by Huang and colleagues. In this review, the authors explored the potential of miRNAs in regulating therapies response in CCA. Notably they evaluated the role endorsed by miRNAs in the response to gold standard CCA treatments (i.e. gemcitabine, 5-FU, cisplatin). A exhaustive list of the oncogenic and tumor suppressor miRNAs described for participating to drug response in CCA is developed in this manuscript. Furthermore, innovative miRNA-based approaches with a great potential for cancer treatment are displayed.
Overall, this study is well referenced and address most of the miRNAs regulating the drug response of CCA. The topic is rather novel in the field and pave the way for innovative therapeutic approaches in CCA. I would recommend this work for publication although the comments below may help to improve the quality of this review.
Point 1: In the introductory section, a few words could be added exhibiting the specificity of CCA microenvironment. Indeed, CCA are associated with a highly desmoplastic stroma which could explain the poor drug response in CCA.
Response 1: Thanks to the reviewer’s suggestion. The prominent features of CCA stromal are desmoplastic and hypovascularized, which are related to the decreased efficacy of anti-tumor drug delivery and CCA progression. We added this part in the introductory section. Please see page 2, line 64-66.
Point 2: As mentioned by the authors, CCA display a molecular heterogeneity discriminating iCCA from eCCA. Notably, these disparities are noticeable in terms of molecular signature, patients survival and response to treatments. Therefore, it should be better specified when possible whether the CCA are intrahepatic or extrahepatic.
Response 2: As the review article covered both intrahepatic and extrahepatic CCA as a whole, we did not specifically divide the sections into intra- and extrahepatic CCA. However, we did specify the tumor samples from intrahepatic or extrahepatic CCA when available in each miRNA section. For example, in 2.10 miR-1249 section, the CCA tissues used for miR-1249 expression is intrahepatic.
Point 3: The authors described mainly results from in vitro studies. In order to complete this review and to go a bit further, the authors could propose CCA clinical trials on adjuvant therapies that could be used in the future to support these experimental results (e.g. the french study (PRODIGE 12—ACCORD 18).
Response 3: Thanks to the reviewer’s suggestion. The review article is focusing on miRNA modulating chemotherapy response and therefore clinical trials of combination of miRNA mimics/inhibitor delivery with standard chemotherapy might be reasonable for going further in the future. However, to date miRNA-based therapeutics combined chemotherapy such as gemcitabine are not convincing in the preclinical development. Thus, we concluded that more advanced technology such as nanomedicine to facilitate the synergistic delivery of chemotherapy and miRNAs. But definitely clinical trials should be encouraged once promising results were demonstrated in preclinical stage.
Point 4: In section 5, the part on miRNAs sponges could be developed. For instance, the type of molecules able to sponge miRNAs could be mentioned (e.g. lncRNAs, circRNAs). Moreover, some of them are already described for their participation in CCA carcinogenesis through miRNA sponging and could therefore be cited here.
Response 4: Thanks to the reviewer for this comment. We added circRNA and lncRNAs as the two common molecules for miRNA sponge. Please see page 13, line 447-448. As miRNA sponge are just an example of anti-miR in this review, we did not mention the role of miRNA sponge in the CCA tumorigenesis. Further review focusing on circular RNA or lncRNA in CCA should include this part.
Point 5: A part could be added on miRNA mediated through extracellular vesicles (EVs) as companion biomarker. Indeed, miRNAs contained in EVs or circulating freely are found in liquid biopsies and have a predictive value for response to therapies.
Response 5: Exosomal miRNA as a biomarker for treatment response is promising and one of the research hotspots recently. We reviewed the literature and only found one study investigating profiles of exosomal miRNA from three CCA cell lines compared with normal cholangiocytes (Life Sci 2018 Oct 1;210:65-75). Future investigations on the association of exosomal miRNAs and chemotherapy response are needed.
[Minor comments]
Point 1: line 14: “improved” should be “improve”
Response 1: We thank to the reviewer. The typo error is corrected. Please see page 1, line 14.
Point 2: line 30-31: according to the current dogma, hilar CCA are included in the extrahepatic bile ducts category.
Response 2: We added the hilar CCA in the extrahepatic CCA category. Please see page 1, line 30-31.
Point 3: line 34: “carries” should be “carry”
Response 3: We changed to “carry”.
Point 4: line 99: “miRNA” should be “miRNAs”
Response 4: We changed to miRNAs in line 109.
Point 5: line 165: “oncogene” should be “oncogenes”
Response 5: We corrected the term in line 189.
Reviewer 4 Report
Revision
The emerging role of microRNAs in regulating the drug response of cholangiocarcinoma
Biomolecules, MDPI
The present article presents a perspective upon CCA therapy resistance and the role of miRNAs in this mechanisms. Although the paper could be of interest for readers focused on this area of study, the proposed chapters are presenting a genralized information without undergoing extensive search and in depth mechanisms. Authros should consider the following.
Authors should keep consistent the use of abreviations and also abreviate the word at its first apereance in the text: eg. microRNAs/miRNAs in the abstract. The same observation is available for the entire manuscript.
Authors should explain in more detail the underling mechanisms presented in Figure 2 as it is an important part of the article. The same is available for Figure 2 and 3
Subchapter 5.1. miRNA-based therapeutics in cancer presents some information that are already known and discuss repeatedly in the literature. Authors should think including more novel strategies regarding miRNAs therapeutics or omit this part.
Subchapter 5.2. The potential role of miRNA-based therapeutics in cholangiocarcinoma is far to superficial and presents only a small number of examples regarding miRNA therapeutics in CCA. Authors should significantly extend this part and divide the studies according to the research protocol – in vitro or in vivo.
Authors also mentioned that miRNAs could function as diagnostic or prognostic markers in CCA. Considering that an early diagnosis strategy would hamper the installation of drug resistance, this subject is of interest. Also, there are miRNAs that could predict the installation of therapy resistance and could have valuable prognostic role; author should also include this part.
The conclusion is very general. Authors are urged to include a more personal look and their opinion about miRNA therapeutics in CCA, including the idea of future perspectives.
Author Response
We thank the reviewer for helpful comments and suggestions. Each issue has been addressed and revisions have been made in the manuscript accordingly. All changes are marked by track changes. Our specific answers to the reviewer’s comments are detailed below.
[Major comments]
The present article presents a perspective upon CCA therapy resistance and the role of miRNAs in this mechanisms. Although the paper could be of interest for readers focused on this area of study, the proposed chapters are presenting a generalized information without undergoing extensive search and in depth mechanisms. Authors should consider the following.
Point 1: Authors should keep consistent the use of abreviations and also abreviate the word at its first appearance in the text: eg. microRNAs/miRNAs in the abstract. The same observation is available for the entire manuscript.
Response 1: We abbreviated the microRNAs to miRNAs in the abstract in concordance with the text (Please see the abstract, page 1 ). In both abstract and text, we indicated microRNAs as miRNAs for the first appearance and then miRNAs for the following context.
Point 2: Authors should explain in more detail the underling mechanisms presented in Figure 2 as it is an important part of the article. The same is available for Figure 2 and 3.
Response 2: Thanks to this reviewer for this comment. We summarized the mechanisms in the figure legends for figure 1 (page 5, line 106-115), figure 2 (page 9, line 300-303), and figure 3 (page 11, line 378-382). Due to the limitation of length for figure legend, we also suggested to read more details in the text.
Point 3: Subchapter 5.1. miRNA-based therapeutics in cancer presents some information that are already known and discuss repeatedly in the literature. Authors should think including more novel strategies regarding miRNAs therapeutics or omit this part.
Response 3: Thanks to this suggestion. We tend to keep this part because the review article is not only for those familiar with miRNA-based therapeutics but also others such as gastrointestinal clinicians, surgeons or oncologists, who might need update information in this field. Therefore, giving a brief summary may facilitate them to capture an overall look of miRNA-based therapeutics.
Point 4: Subchapter 5.2. The potential role of miRNA-based therapeutics in cholangiocarcinoma is far to superficial and presents only a small number of examples regarding miRNA therapeutics in CCA. Authors should significantly extend this part and divide the studies according to the research protocol – in vitro or in vivo.
Response 4: We did the literature review and found no additional studies focusing on delivery of miRNAs to enhance chemotherapy cytotoxicity in cholangiocarcinoma. We are pleased to welcome any studies suggested by the reviewer to add in this section. In the examples mentioned in this section, we clearly indicated whether the results conducted by in vivo experiments. Please see page 13, line 473; page 14, line 485; and page 14, line 494. These studies used both in vitro and in vivo models and it is not feasible to divide them into in vitro or in vivo separately.
Point 5: Authors also mentioned that miRNAs could function as diagnostic or prognostic markers in CCA. Considering that an early diagnosis strategy would hamper the installation of drug resistance, this subject is of interest. Also, there are miRNAs that could predict the installation of therapy resistance and could have valuable prognostic role; author should also include this part.
Response 5: We summarized the specific miRNAs implicated in resistance to chemotherapy. Regarding the role of miRNA in predicting chemotherapy resistance in CCA, we did not find suitable studies that identify a collection of miRNAs from either blood or CCA tumor samples and analyse the correlation with subsequent chemotherapy resistance. We welcomed any specific studies provided by the reviewers to add this part.
Point 6: The conclusion is very general. Authors are urged to include a more personal look and their opinion about miRNA therapeutics in CCA, including the idea of future perspectives.
Response 6: Thanks to the reviewer’s suggestion. However, we tend to state the conclusion according to the updated scientific evidence. The efficacy of miRNA delivery and interaction with tumor microenvironment are complicated. Therefore we need more experimental research using advanced biotechnology before drawing any specific conclusion, which is our opinion.
Round 2
Reviewer 1 Report
In my opinion, the article is suitable for publication in the present form.
Reviewer 4 Report
Accept in the present form